# DFT and TD-DFT Investigations for the Limitations of Lengthening the Polyene Bridge between N,N-dimethylanilino Donor and Dicyanovinyl Acceptor Molecules as a D-π-A Dye-Sensitized Solar Cell

**DOI:** 10.3390/ijms25115586

**Published:** 2024-05-21

**Authors:** Sharif Abu Alrub, Ahmed I. Ali, Rageh K. Hussein, Suzan K. Alghamdi, Sally A. Eladly

**Affiliations:** 1Department of Physics, College of Science, Imam Mohammad Ibn Saud Islamic University (IMSIU), Riyadh 11623, Saudi Arabia; snabualrub@imamu.edu.sa; 2Basic Science Department, Faculty of Technology and Education, Helwan University, Saraya El Koba, El Sawah Street, Cairo 11281, Egypt; 3Department of Applied Physics, Kyung Hee University, Yongin 17104, Republic of Korea; 4Physics Department, Faculty of Science, Taibah University, Madinah 44256, Saudi Arabia; skghamdi@taibahu.edu.sa; 5Basic Science Department, Modern Academy of Engineering and Technology, Cairo 11439, Egypt; saly.aziz@eng.modern-academy.edu.eg

**Keywords:** polyene bridge, D-π-A dye, DFT, TD–DFT, UV–Vis spectrum, photovoltaic properties

## Abstract

One useful technique for increasing the efficiency of organic dye-sensitized solar cells (DSSCs) is to extend the π-conjugated bridges between the donor (D) and the acceptor (A) units. The present study used the DFT and TD–DFT techniques to investigate the effect of lengthening the polyene bridge between the donor N, N-dimethyl-anilino and the acceptor dicyanovinyl. The results of the calculated key properties were not all in line with expectations. Planar structure was associated with increasing the π-conjugation linker, implying efficient electron transfer from the donor to the acceptor. A smaller energy gap, greater oscillator strength values, and red-shifted electronic absorption were also observed when the number of polyene units was increased. However, some results indicated that the potential of the stated dyes to operate as effective dye-sensitized solar cells is limited when the polyene bridge is extended. Increasing the polyene units causes the HOMO level to rise until it exceeds the redox potential of the electrolyte, which delays regeneration and impedes the electron transport cycle from being completed. As the number of conjugated units increases, the terminal lobes of HOMO and LUMO continue to shrink, which affects the ease of intramolecular charge transfer within the dyes. Smaller polyene chain lengths yielded the most favorable results when evaluating the efficiency of electron injection and regeneration. This means that the charge transfer mechanism between the conduction band of the semiconductor and the electrolyte is not improved by extending the polyene bridge. The open circuit voltage (V_OC_) was reduced from 1.23 to 0.70 V. Similarly, the excited-state duration (τ) decreased from 1.71 to 1.23 ns as the number of polyene units increased from n = 1 to n = 10. These findings are incompatible with the power conversion efficiency requirements of DSSCs. Therefore, the elongation of the polyene bridge in such D-π-A configurations rules out its application in solar cell devices.

## 1. Introduction

In order to satisfy the growing demand for clean energy sources worldwide, it is becoming increasingly essential to explore and develop sustainable and renewable energy sources. Dye-sensitized solar cells, or DSSCs, are attracting more attention from researchers due to their potential as a cheaper and more environmentally friendly renewable energy source [1,2,3]. DSSCs are generally constructed from a semiconductor layer, a redox electrolyte, and a dye molecule that serves as the light-capturing core. Building DSSCs with TiO_2_ as the semiconductor surface has many advantages, including high efficiency, low cost, and ease of manufacturing [4,5]. Basically, dyes or molecules are considered to be the primary component of DSSCs as they transform absorbed solar energy into electricity by initiating charge separation and transfer processes [6,7,8,9]. The main structural property of the dyes is their structural design D-π-A, in which D and A are the donor and acceptor parts, respectively. The π-linker connecting the dye’s D and A parts enables the intramolecular transfer of charge between them. The performance and effectiveness of DSSCs are heavily based on the proper choice of these dye molecules. There has been a significantly large number of studies conducted on organic and inorganic dyes to draw conclusions about how their structural, electronic, and spectroscopic properties relate to the desired performance efficiency of DSSCs [10,11,12,13,14]. The usage of substituted triphenylamine (TPA) as a donor component and π-spacers with a thiophene ring has shown a favorable framework for the production of efficient organic dyes in DSSCs [15]. Several acceptor components have been represented by chemical substituents that can serve as an anchor to facilitate their adsorption onto a metal oxide surface, which in turn activates electron injection. Carboxylic acid and cyano-acrylic acid groups were the most commonly used anchor components as acceptor parts in DSSCs [16,17]. Novel anchor groups have surfaced in recent years, expanding the range of materials that can be used to create dye-sensitive solar cell dyes. Their corresponding physical and chemical properties provide noteworthy effects at the dye–metal oxide interface [18,19,20,21]. One of the most effective strategies to improve intermolecular interactions and hence device performance is to modify the structure of the core π-bridge by lengthening the π-conjugation units [22]. The synthesis and characterization of a series of organic dyes with elongating conjugated bridges have demonstrated that these dyes progressively produce red-shifted electronic absorption spectra and a gradual reduction in energy gap [23,24,25,26]. It has proven convenient to control the band gap of a series of chromophores by varying the length of the spacers connecting 4-(dimethyl-amino) anilino donors and C(CN)2 acceptors [27]. The optical, photovoltaic, and electrochemical characteristics performance of oligothiophenes with a 4,8-bis(thienyl)benzo[1,2-b:4,5-b′] di-thiophene core was shown to be strongly affected by the length of the π-conjugation bridges [15]. Density functional theory (DFT) represents a theoretical foundation that provides an accurate means of predicting and understanding the behavior of materials in optoelectronic applications. Moreover, this approach enables researchers to investigate and comprehend semiconductors and complex molecules, as well as their associated interactions [28]. The properties of the different dye types used in the application of DSSCs have been the subject of an excessive number of studies employing theoretical techniques These studies focused on clarifying how the desirable properties, such as intramolecular charge transfer (ICT), wavelength of light absorbed, and photovoltaic parameters, of the dye molecules, could be correlated with their molecular geometries, electronic structure, and spectroscopic characteristics [29,30,31]. 

In this study, the role of lengthening the polyene molecule as a conjugated bridge in the D-π-A dye structure between donor N,N-dimethylanilino (DMA), and acceptor dicyanovinyl (DCV) has been investigated by using DFT and TD–DFT techniques. The DFT investigation of the selected dyes enabled the determination of certain properties, such as ideal molecular geometry and some electronic structure parameters. The optical properties of the sensitizers were calculated and analyzed using the TD–DFT method based on maximum absorption, light harvesting efficiency. This work aimed to provide insight into how the number of conjugated polyene units affects the intrinsic mechanisms for DSSC performance design and evaluation.

## 2. Results and Discussion

### 2.1. Optimized Molecular Structure

The dihedral angles between the acceptor DCV, successive polyene units, and the donor DMA were calculated and listed in Appendix A. Figure 1 utilizes the dye n = 2 as an example to demonstrate the dihedral angle of concern between its planes. It can be seen that the skeletal structures along the DMA–polyene bridge and DCV exhibit quasi-planar structures. For n = 1 to n = 10, the dihedral angle φ D, which represents the angle between the constituent fragments of the donor DMA, alters between −179.99 and 179.99°. Dihedral angles varying between 179.99, −179.99, 180.00, and −180.00° are found between the donor unit and the polyene chain (φ 0), as well as between the polyene chain and the acceptor unit (φ A). The dihedral angles between the successive polyene planes also fall within the mentioned range. These results indicate a planar structure of the conjugation within the polyene bridge as well as between the donor and acceptor fragments across the bridge, which strongly supports the intramolecular charge transfer mechanism. The compounds with n = 11 to n = 13 continued in a similar manner, as demonstrated in Appendix A, which means that the dihedral angle values were no different from those of their predecessors. Therefore, increasing the number of polyene spacers maintains the planar structure of the dyes and does not affect the charge transfer from donor to acceptor units. 

### 2.2. Electronic Properties

A key measure of the electron transfer capabilities of the dyes are the frontier molecular orbitals, commonly referred to as the highest occupied molecular orbital (HOMO) and the lowest unoccupied molecular orbital (LUMO). The bandgap energy (Eg, difference between HOMO and LUMO energy levels), ionization potential (IP = −E_HOMO_), and electron affinity (EA = −E_LUMO_) were calculated and recorded in Table 1. These parameters obtained from Koopmans’ theorem [32] are usually employed as primary indicators for evaluating the chemical reactivity of dye sensitizers. It is obvious that increasing the conjugated polyene bridge reduces gap energies Eg, which basically promotes exciton generation and improves photoexcitation efficiency. It is important to keep in mind that standard density functional theory techniques commonly underestimate band gaps. This could be due to an energy derivative discontinuity with respect to electron number or a convergence difficulty with the selected basis set, which results in errors, especially towards LUMO energies [33]. Moreover, increasing the conjugated units causes an increase in electron affinity and a decrease in the ionization potential. This result is regarded as having good donor and acceptor properties for the sensitizer dye, since high electron affinity (EA) and low ionization potential (IP) provide both good donor and acceptor properties, according to the full-electron donor-acceptor map (FEDAM) by Martínez et al. [34].

An effective dye has requirements that must be satisfied by its levels of HOMO and LUMO energy. The LUMO level should be above the conduction band of TiO_2_ (−4.0 eV) to potentially induce electron injection from the excited state of the dye onto the TiO_2_ surface. Dye regeneration is also feasible after electron transfer if the HOMO level of the dye lies below the redox potential of the electrolyte I^−^/I^3−^ (−4.8 eV). Figure 2 illustrates the arrangement of the HOMO and LUMO energy levels for the studied dyes. Based on this figure, it can be noticed that every dye has an E_LUMO_ that is high enough to be energetically advantageous for electron injection into the conduction band of TiO_2_. The dye regeneration procedure will also be successful since the HOMO levels are located below the redox potential. The HOMO and LUMO energy levels, however, considerably shift when the conjugated bridge is increased. As seen in the figure, increasing the conjugations lowers the LUMO and raises the HOMO level. The LUMO level dropped by 0.53 eV from n = 1 to n = 10, and the HOMO level rose dramatically (by 0.89 eV) for the same order. The performance of the DSSCs is negatively impacted by raising the HOMO since it lowers the open-circuit voltage [35]. On the other hand, as Appendix A shows, the HOMO level for n = 11, 12 and 13 was −4.79 and −4.75 and −4.72 eV, respectively. Consequently, increasing the polyene bridge above n = 10 will cause HOMO levels to rise above the redox potential, impeding the process of regeneration. Thus, the studied material will function effectively up to the polyene bridge length limit of n = 10. 

HOMO and LUMO frontier molecular orbitals are considered to be crucial factors for studying the electronic properties of any materials. Electron-donating regions are known as HOMO, and electron-accepting regions are known as LUMO. The HOMO and LUMO graphs of dyes with n = 1 and n = 10 are displayed in Figure 3. The HOMO consists of primary bonding orbitals, which are distributed on the benzene ring of the DMA and the C=C of the polyene bridge. HOMO also include antibonding orbitals distributed on N(CH_3_)_2_ of the DMA and the DVC unit. LUMO are bonding orbitals that are extended on the C-C bond of the polyene bridge. The DVC contributes to LUMO through a large lobe on the carbon atom and two lobes that are located on the two cyanide groups. As can be seen in the figures, the complementarity and overlap of HOMO and LUMO on the conjugated bridge facilitate intramolecular charge transfer from donor to acceptor units. Furthermore, the ability to separate HOMO and LUMO is expected to facilitate the intramolecular charge transfer from donor to acceptor, which is then injected into the thin film surface. Observation of the HOMO and LUMO figures is noteworthy. As the number of conjugated units increases on the polyene chain, the most distant HOMO lobes from the DMA unit become smaller. Similarly, with LUMO, the furthest LUMO lobes from the DVC unit become smaller as n increases. This is most apparent when n = 11, 12, and 13, as seen in Appendix A. This certainly has a detrimental effect on intramolecular charge transfer within the studied dyes. Therefore, we can conclude that, with fewer polyene units, the intramolecular charge transfer from donor to acceptor units is more efficient.

### 2.3. Absorption Spectra

The absorption spectra simulated by the CAM-B3LYP/6-311G (d, p) theoretical model for the ten dyes are displayed in Figure 4. A notably broad and maximized absorption peak was produced by extending the conjugated bridge from n = 1 to n = 10, which improved the light-harvesting abilities of the dye sensitizer. The electronic absorption spectra are gradually redshifted when the polyene spacer is increased. The observed red-shifted absorption can potentially be attributed to the better electron-donating abilities of DMA and C=C units. Table 2 provides a summary of the excitation energy, maximum wavelength, oscillator strength, and electronic transitions. As shown in the table, the range where the strongest UV–vis absorption maxima for all 10 dyes are located lies within 450–660 nm. The absorption spectrum of dyes covers the majority of the solar spectrum and, hence, maximum solar harvesting is expected. 

The main electron transition contribution is attributed to the HOMO→LUMO transition, as Table 2 shows. This is confirmed by Figure 4, which indicates that the strongest absorption peak in each spectrum is the peak with the longest wavelength. The HOMO to LUMO transition, or the type transition of the lowest singlet electronic excitation, corresponds to the π to π* transitions. The high oscillator strength value has a positive correlation with high absorbance light efficiency. The electronic transitions are significantly associated with high oscillator strength values (up to 5.29 for n = 10), indicating that the dyes are strongly harvesting light. Dyes n = 11 to n = 13 behave in the same way as n < 10, as can be seen in Appendix A. The absorption spectra continue to broaden, maximize, and redshift. For n = 13, the oscillator strength reached an extremely high value of 6.52, indicating high efficiency of light harvesting. It follows that the absorption properties are improved by increasing the polyene spacer without having any negative effects.

The effect of solvent polarity on the photophysical properties was investigated by using the same level of theory (CAM-B3LYP/6-311G (d, p)) to calculate the absorption spectra in toluene and acetonitrile solvents. Solvent polarity impact is shown by the absorption wavelength peaks for toluene and acetonitrile given in Appendix A, and Appendix A, respectively. Compared to the chloroform solvent, a very low blue shift is observed when the solvent changes from chloroform to toluene and finally to acetonitrile. Toluene and acetonitrile maintain the basic properties provided by the chloroform solvent without any notable difference. The conjugated bridge produced a broad and maximum absorption peak as it extended from n = 1 to n = 10.

### 2.4. Photovoltaic Properties

The free energy for electron injection ΔG ^inject^ can be utilized for describing the electron injection from the excited state of the dyes to the conduction band of TiO_2_, which determines the electron injection rate and, consequently, the current density JSC in DSSCs. The driving force behind the regulation of electron collection and dye regeneration yield is the potential difference between the oxidized dyes and the electrolyte. This difference must be sufficiently large to give a sufficient regeneration driving force ΔG ^reg^ for efficient regeneration of the dye’s ground state. Table 3 provides the calculated values of ΔG ^inject^ and ΔG ^reg^ for the studied dyes (n = 1 to 10). In order to ensure efficient dye performance, the absolute values of ΔG ^inject^ and ΔG ^reg^ have to exceed 0.2 eV [36]. ΔG ^inject^ satisfies this requirement by having absolute values between 0.94 and 1.11 eV, while the higher number of polyene units could delay the charge regeneration process, since ΔG ^reg^ values are less than 0.2 eV. A higher driving force for electrons injected into the semiconductor surface from the excited states of the dyes is indicated by a higher negative value of ΔG ^inject^ [37]. The dye n = 1 has the most negatively valued injection energy, which suggests a faster rate of electron injection. The negativity of the ΔG ^inject^ has been shown to decrease from −1.22 eV (n = 1) to −0.94 eV (n = 8), after which it then begins to increase to −1.00 eV (n = 9) and −1.01 eV (n = 10). The negativity then starts to rise for n = 11, 12, and 13 (as clarified in Appendix A) at an extremely slow rate of just 0.01 eV. Consequently, the ΔG ^inject^ value is unaffected when incorporating more polyene units after n = 10. In other words, increasing the polyene units will not enhance the ΔG ^inject^ value, or rather the J SC, and the best value was obtained at n = 1. 

A high ability of the dye to recover electrons from the electrolyte and subsequently increase photoelectric conversion efficiency corresponds to a high ΔG ^reg^ value [38]. The calculated reorganization energy values exhibited a consistent decrease as the polyene spacer increased. Thus, dyes with fewer polyene units (smaller n) will have effective electron regeneration, leading to higher power conversion efficiency.

LHE is the percentage of photons in the incident light that the dye absorbs and uses to generate an electron-hole pair. Thus, the capacity of the dye to absorb light and convert it into an excited state is measured by the LHE value. According to the obtained results shown in Table 4, we find that the insertion of polyene units increases the LHE value since LHE depends only on the oscillator strength [39]. That is, the power conversion efficiency is improved by increasing the number of polyene spacers within the dye. For n > 10, the behavior of the results has not changed, and LHE keeps increasing until it attains its maximum value at n = 13 (as shown in Appendix A). The voltage across the DSSC terminals without any kind of electrical current is represented by the open-circuit voltage (V_OC_), which is the maximum voltage the cell can produce when there is no external load or current passing through it. V_OC_ is directly related to the difference between the E LUMO of the dye and the conduction band of the semiconductor, as shown by Equation (4). This implies that a higher E_LUMO_ level will result in a higher V_OC_. As previously understood, the LUMO level decreases towards TiO_2_ CB when the polyene spacer is increased. This is consistent with the values shown in Table 4, where the V_OC_ value decreases starting at n = 1 and reaches its lowest value of 0.70 V at n = 10. One important factor affecting charge transfer efficiency is the excited-state lifetime. The excited-state lifetime is the duration of time it takes for an excited dye molecule to revert to its ground state. A longer excited-state lifetime increases the probability of long-term stability and is expected to facilitate charge transfer [40]. Table 4 indicates that the excited-state lifetime of n = 1 is the highest and that it decreases in sequence from n = 1 to n = 10. This was confirmed when the polyene group increased at n = 11, 12, and 13, which resulted in a continuous decrease in the lifetime value, as shown in Appendix A. This indicates that the addition of the polyene spacer group has an adverse impact on the excited-state lifetime and, consequently, on the performance of the device.

The impact of extending the π-spacer on the photovoltaic performance of D-π-A organic dyes has been the subject of numerous previous theoretical investigations. All of these studies agreed that increasing the π-spacer can improve the opto-electronic properties of the dyes being studied, as was pointed out in the introduction. The bulk of the current findings, it could be argued, diverge from those of earlier studies. Increasing the polyene chain did, in fact, succeed in improving the UV–vis absorption properties and reducing the energy gap, in line with those findings. On the other hand, the most important electronic and photovoltaic properties were found to contradict the familiar results of those studies. The primary indicator of this was the rise in the HOMO level above the TiO_2_ CB caused by increasing the conjugated spacer, which is likely to cause inconsistencies regarding the dye’s mechanism of action in the solar cell. Improved ΔG ^inject^ and ΔG ^reg^ values, as well as longer excited-state lifetime, were obtained in previous studies by extending the π spacer; however, in the current study, increasing the polyene was not beneficial for those quantities and produced a shorter excited-state lifetime. 

The broad implications of the work suggest that long-chain polyene polymers are not recommended to be used as the core for building an effective D-π-A dye. The efficient performance of the current D-π-A configuration is constrained to shorter polyene-conjugated bridges. It has been demonstrated in numerous earlier studies that polyene combined with other organic dyes has positive effects on photovoltaic qualities for extremely short polyene lengths without being subjected to the effects of lengthening the polyene bridge [41,42]. This could be due to the nature of the linear polyene chain itself, or maybe other types of donors or acceptors would produce different results. Finally, this motivates further study to provide a better understanding of the design and synthesis of organic dyes for DSSCs that incorporate the polyene chain.

## 3. Materials and Methods

### 3.1. Chemistry and Molecular Design 

Most recent studies have focused on material types with donor and acceptor units connected by π-conjugated bridges as a prospective technique to enhance DSSC performance. One of these types used N,N-dimethylanilino (DMA) as the donor, connected by a polyene bridge to dicyanovinyl (DCV) as the acceptor group. This structural design represents the D-π-A configuration of organic dye sensitizers. The molecular structure of the polymer, including the DMA and DVC components, is shown in Figure 5. Bogdanov and his colleagues have recently reported the synthesis of n = 2 and 3, namely 2-{(2E,4E)-5-[4-(dimethyl-amino) phenyl] penta-2,4-dien-1-ylidene} malononitrile and 2-{(2E,4E,6E)-7-[4-(dimethyl-amino) phenyl] hepta-2,4,6-trien-1-ylidene} malononitrile [43]. The two precursors involved in the synthesis that is being described are an extended aromatic aldehyde and malononitrile. Knovenagel condensation was used to produce the final product, which had the donor and acceptor connected by the π-conjugated bridge. Ten dye molecules will basically be the subject of the current work, i.e., from n = 1 to n = 10. The effect of expanding the polyene with a higher order (n = 11–13) was addressed in the Appendix A. Since they have a low molecular weight, organic dyes have poor light and chemical stability, with an especially negative effect on their thermal stability. The development of DSSCs using conjugated polymer thin films has grown substantially in recent years as a result of their many advantages. The high density of the polymer chain, which provides better thermal stability than small molecules, is one of these advantages. The alternating double and single bonds in the conjugated polymer result in stronger bonds with a higher molecular weight. Polythiophene (PT) is one of the most widely studied conjugated polymers due to its facile synthesis and excellent thermal stability. These features give it a wide range of electronic uses, including as chemical sensors, organic solar cells, and organic field-effect transistors [44]. Energy reduction and molecular stabilization may be achieved using a conjugated π bond system formed by alternating single and double bonds within the polyene polymer. The proposed dyes could therefore possess a high durability potential because of the stability of conjugated polyene polymers.

### 3.2. Computational Methods

The ground-state optimized geometries and electronic properties of all dyes were calculated using the B3LYP hybrid functional level, which emerged from the DFT method as having the best geometry optimization performance [45,46]. The basis set type employed was 6-311G (d, p), and the theoretical model was referred to as “B3LYP/6-311G (d, p)”. The given model was chosen after evaluating several well-known functionals (B3LYP, ωB97XD, and MPW1PW91), which were used to optimize the selected dyes using the same basis set 6-311G (d, p). The UV–vis absorption wavelength was then calculated using TD–DFT so that we could determine the method for optimization that produced the result that was most consistent with the experimental value reported in the literature. The B3LYP was selected since it provided the most compatible results with the experimental value, as indicated in Appendix A. The TD–DFT has been shown to accurately predict molecular charge-transfer spectra by using the Coulomb-attenuating model CAM-B3LYP functional to overcome B3LYP limitations [47,48,49]. Therefore, the CAM-B3LYP/6-311G (d, p) model was used for calculating the excitation energy (E ex), absorption wavelengths (λ max), oscillator strength (f), and UV–vis absorption spectra of all dyes in chloroform solvent. The computational chemistry program Gaussian 09 was used for all of the calculations given in the work [50]. All the results were extracted and visualized using Avogadro software 1. 2. 0 and the Chemcraft 1.8 graphical program [51,52].

Dispersion correction is essential to the synthesis, stability, and functionality of materials, especially for those systems that have a large number of conjugated double bonds. Thus, DFT calculations require long-range correlation, since it is the physical foundation of dispersion interaction. Long-range correction in exchange functionals means increasing the long-range exchange effect by replacing the Hartree–Fock exchange integral with the long-range component of the exchange functional. The underestimating of oscillator strengths and charge transfer excitation energies, as well as improvement in poor optical response in time-dependent calculations, have been resolved by long-range correction. Therefore, it is believed that applying the long-range correction increases the precision of quantum chemical calculations [53].

### 3.3. Photovoltaic Parameters

The energy conversion efficiency of any solar cell type is substantially affected by specific factors. The two most relevant of these parameters are short circuit current density (JSC) and open-circuit photovoltage (V_OC_). One important component in the calculation of J SC is light harvesting efficiency (LHE), which is a function of the oscillator strength (f) of the absorbed dye molecule. Electronic injection-free energy (ΔG ^inject^), regeneration driving force (ΔG ^reg^), and excited-state lifespan (τ) are also highly helpful measures in photovoltaic performance. The following equations provide a mathematical description of each of those parameters [54,55,56,57], which will be discussed in more detail in the results section.
$E^{dye}=-E_{HOMO}$(1)$E^{dye}$ is the ground-state oxidation potential of dye $E^{dye*}=E^{dye}-E_{ex}$(2)$E^{dye*}$ is the oxidation potential of the excited dye $E_{ex}$ is the lowest excitation energy that corresponds to λ_max_$\Delta G^{inject}=E^{dye*}-\left|E_{CB}\right|$(3)$E_{CB}$ is the reduction potential of the conduction band edge of TiO_2_, (−4.0 eV)$\Delta G^{reg}=\left|E_{redox}\right|-E^{dye}$(4)$E_{redox}$ is the redox potential I^−^/I^3−^ (−4.8 V)$V_{oc}=E_{LUMO}-E_{CB}$(5)
$LHE=1-10^{-f}$(6)
$V_{OC}=E_{LUMO}-E_{CB}$(7)
$\tau =\frac{1.499}{fE_{ex}^{2}}$(8)


## 4. Conclusions

This paper investigated D-π-A dyes, which consist of polyene as a π-conjugated bridge linking the DMA donor molecule and DCV acceptor unit, using the DFT and TD-DFT techniques. The objective was to study the effect of lengthening the polyene linker on the electronic and optical properties of the designed dyes. Even though the extension of the polyene bridge resulted in several advantageous properties, such as reduced energy gab, enhanced oscillator strength, and red-shifted absorption peaks, there were some unfavorable consequences that would imply poor dye performance. With the expansion of the polyene bridge, the HOMO energy levels increased at a notable rate, even becoming higher than the redox potential of the electrolyte, which represents a major barrier to energy conversion efficiency for the cell. On the extended skeleton of higher polyene units, electron transfer may be affected by the shrinking of some HOMO and LUMO orbitals. The calculated injection and regeneration energies showed that dyes with fewer polyene units are able to achieve fast dye regeneration as well as effective interface charge injection. Adding a polyene spacer in the core slightly improved the light-harvesting efficiency (LHE), but the efficiency of electron injection from the E_LUMO_ of the dye into the TiO_2_ CB was declined by sloping values of open-circuit voltage (V_OC_) and excited-state lifetime (τ). In summary, lengthening the polyene polymer as the core bridge between the donor DMA and the acceptor DCV is not an effective strategy to build DSSCs due to its limited capability to enhance the efficiency of energy conversion or device performance.

## Figures and Tables

**Figure 1 ijms-25-05586-f001:**
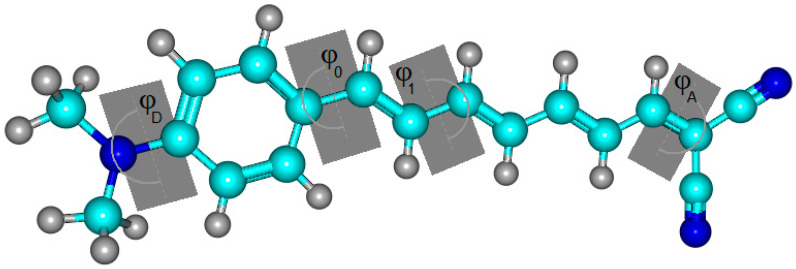
The dihedral angles for dye (n = 2) demonstrate the dihedral angle between the dye molecules’ planes.

**Figure 2 ijms-25-05586-f002:**
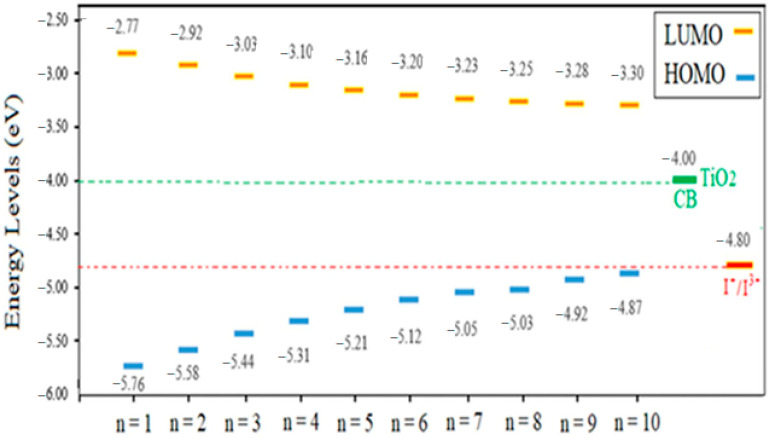
Diagram showing the HOMO and LUMO energy levels for dyes with n = 1 to n = 10.

**Figure 3 ijms-25-05586-f003:**
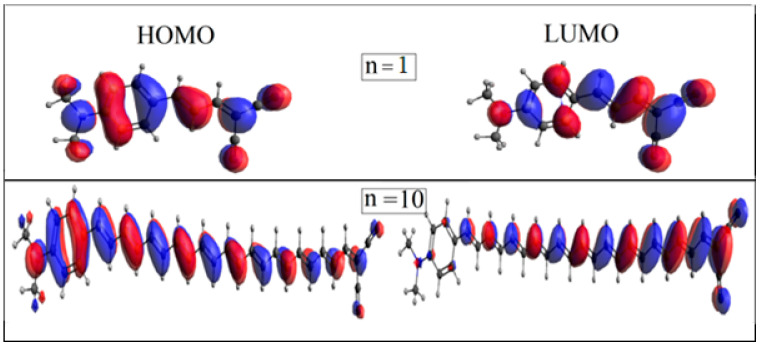
HOMO and LUMO frontier molecular orbital distribution of dyes n = 1 and n = 10.

**Figure 4 ijms-25-05586-f004:**
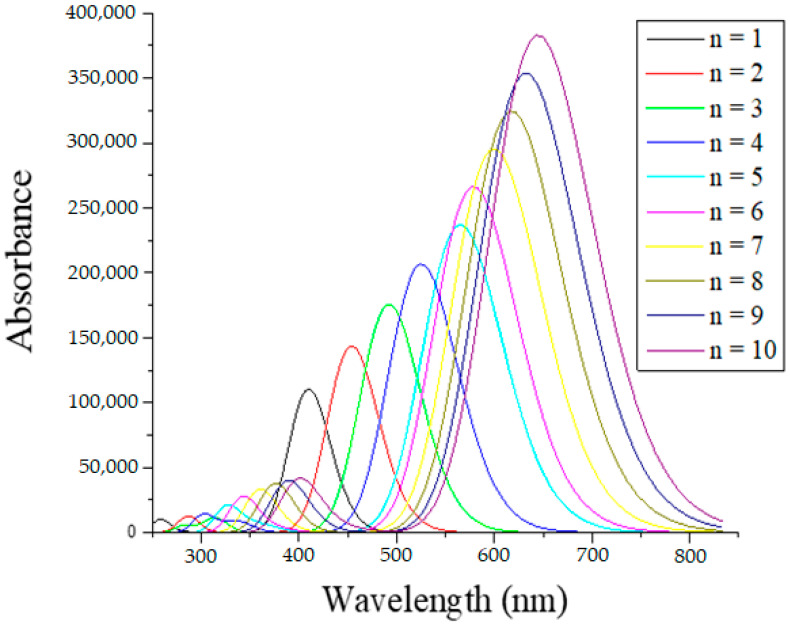
The absorption spectrum of dyes n = 1 to n = 10 simulated by the TD-CAM-B3LYP/6-311G (d, p) method in chloroform solvent.

**Figure 5 ijms-25-05586-f005:**
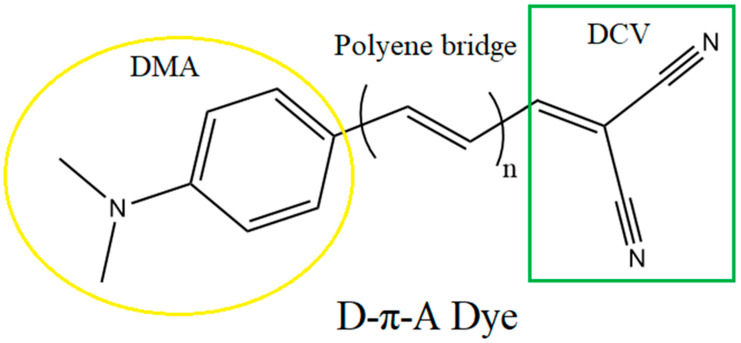
The studied molecule included a polyene bridge connecting the donor unit, DMA, to the acceptor unit, DCV, with n = 1, 2, ….

**Table 1 ijms-25-05586-t001:** The band gap (Eg), ionization potential (IP), and electron affinity (EA) for all studied dyes, calculated by DFT/B3LYP at 6-311G (d, p) model.

Compounds	Eg (eV)	IP (eV)	EA (eV)
n = 1	2.99	5.76	2.77
n = 2	2.66	5.58	2.92
n = 3	2.41	5.44	3.03
n = 4	2.21	5.31	3.10
n = 5	2.05	5.21	3.16
n = 6	1.92	5.12	3.20
n = 7	1.82	5.05	3.23
n = 8	1.78	5.03	3.25
n = 9	1.65	4.92	3.28
n = 10	1.58	4.87	3.30

**Table 2 ijms-25-05586-t002:** Electronic transition properties of all dyes calculated at TD–DFT-CAM-B3LYP/6-311G (d, p).

Compounds	E _eX_ (eV)	Wavelength (nm)	Oscillator Strength (ƒ)	Transition	Major Contribution
n = 1	2.98	415.594	1.52	HOMO→LUMO	99%
n = 2	2.69	461.024	1.98	HOMO→LUMO	94%
n = 3	2.48	500.477	2.43	HOMO→LUMO	92%
n = 4	2.32	534.918	2.86	HOMO→LUMO	90%
n = 5	2.20	564.767	3.27	HOMO→LUMO	87%
n = 6	2.10	590.397	3.68	HOMO→LUMO	85%
n = 7	2.06	612.294	4.08	HOMO→LUMO	82%
n = 8	1.97	630.830	4.49	HOMO→LUMO	79%
n = 9	1.92	646.487	4.89	HOMO→LUMO	75%
n = 10	1.88	659.626	5.29	HOMO→LUMO	71%

**Table 3 ijms-25-05586-t003:** Data describing the free energy change for dye regeneration and electron injection for dye from n = 1 to n = 10.

Compounds	E _eX_ (eV)	E ^dyes^ (eV)	E ^dyes^* (eV)	ΔG ^inject^ (eV)	ΔG ^reg^ (eV)
n = 1	2.98	5.76	3.03	−1.22	0.96
n = 2	2.69	5.58	2.89	−1.11	0.78
n = 3	2.48	5.44	2.96	−1.04	0.64
n = 4	2.32	5.31	2.99	−1.01	0.51
n = 5	2.20	5.21	3.02	−0.99	0.41
n = 6	2.10	5.12	3.02	−0.98	0.32
n = 7	2.03	5.05	3.03	−0.98	0.25
n = 8	1.97	5.03	3.06	−0.94	0.23
n = 9	1.91	4.92	3.00	−1.00	0.12
n = 10	1.88	4.87	2.99	−1.01	0.07

**Table 4 ijms-25-05586-t004:** The calculated light harvesting efficiency (LHE), open-circuit photovoltage (VOC), and excited-state lifetime (t) for dyes n = 1 to n = 10.

Compounds	LHE	V_OC_ (eV)	τ (ns)
n = 1	0.9698	1.23	1.71
n = 2	0.989529	1.08	1.61
n = 3	0.996285	0.97	1.54
n = 4	0.99862	0.9	1.50
n = 5	0.999463	0.84	1.46
n = 6	0.999791	0.8	1.42
n = 7	0.999917	0.77	1.33
n = 8	0.999968	0.75	1.32
n = 9	0.999987	0.72	1.28
n = 10	0.999995	0.70	1.23

## Data Availability

The original contributions presented in the study are included in the article, further inquiries can be directed to the corresponding author.

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
