# Peer review of "DFT and TD-DFT Investigations for the Limitations of Lengthening the Polyene Bridge between N,N-dimethylanilino Donor and Dicyanovinyl Acceptor Molecules as a D-π-A Dye-Sensitized Solar Cell"

_ijms, 2024, doi:10.3390/ijms25115586_

Round 1

Reviewer 1 Report

Comments and Suggestions for Authors

The authors investigate the effect of extending the π-conjugated polyene bridge in D-π-A structured dye molecules on the photovoltaic properties of dye-sensitized solar cells (DSSCs). Using DFT and TD-DFT computational methods, the study provides insights into the electronic and optical behaviors of these dye molecules as the length of the polyene bridge varies. The findings suggest that while certain photophysical properties improve with an increase in the conjugation length, such as the oscillator strength and red-shift in absorption spectra, critical aspects like the HOMO energy levels and electron life times present challenges that may hinder the practical application of these dyes in solar cells. However, there are some aspects where revisions are recommended:

1.      The manuscript could benefit from a more detailed explanation of the choice of computational methods and basis sets. Why were B3LYP and CAM-B3LYP chosen specifically for this study? A justification based on comparative performance with other functionals could strengthen this choice.

2.      The study mentions calculations in chloroform solvent; however, the impact of solvent on the photophysical properties of the dyes is not discussed in detail. Could the authors elaborate on how different solvents might affect the results?

3.      While the manuscript cites relevant literature, it lacks a direct comparison between this work's findings and those of similar studies. Can the authors discuss how their results align or differ from existing studies on D-π-A structured dyes in DSSCs?

4.      The manuscript discusses extending the polyene bridge from n = 1 to n = 13. It would be beneficial to discuss the synthetic feasibility and stability of these extended systems, especially in practical applications.

5.      The broader implications of the study for the design of new DSSCs are hinted at but not fully explored. Can the authors discuss potential design rules or guidelines for creating more effective D-π-A dyes based on their findings?

Author Response

Response to Reviewer #1's comments

(Manuscript: ijms-3011105)

We are very much thankful to the reviewer for his deep and thorough review. We have revised our present research paper in light of his useful suggestions and comments. We hope our revision has improved the paper to a level of satisfaction. Number-wise answers to his specific comments and suggestions are as follows:

Comment 1;

- The manuscript could benefit from a more detailed explanation of the choice of computational methods and basis sets. Why were B3LYP and CAM-B3LYP chosen specifically for this study? A justification based on comparative performance with other functionals could strengthen this choice.

Response; 

Thank you for the note; A justification for the choice of B3LYP and CAM-B3LYP was added based on comparative performance with other functionals. The WB97XD and MPW1PW91 were used to optimize the structure of selected dyes. The maximum absorption wavelength was then calculated by TD-DFT and compared with the literature. To illustrate these findings, the dye n = 2 result was used (Table S1). (Lines 141 to 147)

Comment 2;

- The study mentions calculations in chloroform solvent; however, the impact of solvent on the photophysical properties of the dyes is not discussed in detail. Could the authors elaborate on how different solvents might affect the results?

Response;

Investigating how solvent polarity affects the photophysical properties was done in accordance with the reviewer's suggestion. The corresponding results from the calculation of the absorption spectra in toluene and acetonitrile solvents for the studied dyes were recorded in Figures S4, S5, and Tables S7 and S8. (Lines 294 to 302)

Comment 3;

While the manuscript cites relevant literature, it lacks a direct comparison between this work's findings and those of similar studies. Can the authors discuss how their results align or differ from existing studies on D-π-A structured dyes in DSSCs?

Response;

Based on the reviewer's advice, A comparison between the results of this work and those of related studies was discussed, and the key findings that differ from the previous studies on D-π-A structured dyes in DSSCs were included. (Lines 359 to 371).

Comment 4;

The manuscript discusses extending the polyene bridge from n = 1 to n = 13. It would be beneficial to discuss the synthetic and stability of these extended systems, especially in practical applications.

Response;

As per the reviewer's recommendation, the synthetic and stability of the extended systems, have discussed. (Lines 120 to 132)

Comment 5;

The broader implications of the study for the design of new DSSCs are hinted at but not fully explored. Can the authors discuss potential design rules or guidelines for creating more effective D-π-A dyes based on their findings?

Response;

As suggested by the reviewer, a discussion of the potential design rules for creating more effective D-π-A dyes based on the current study has been added. (Lines 372 to 380)

Thanks in Advance

Reviewer 2 Report

Comments and Suggestions for Authors

In this work, the authors studied extended π-conjugated bridges between donor and acceptor units in organic dye-sensitized solar cells (DSSCs) using DFT and TDDFT. This work shows some properties such as increased π-conjugation and red-shifted electronic absorption improve, limitations arise as the HOMO level rises beyond the redox potential of the electrolyte. Smaller polyene chain lengths yield better efficiency in electron injection and regeneration, contrasting with the power conversion efficiency requirements of DSSCs.

This manuscript can be improved before the publication. I would like to ask the authors to consider the comments below.

1. page 1, title

Typo. “DF-DFT” should be “TD-DFT”.

2. page 3, computational details

Some references are missing. For the B3LYP functional: Lee, Chengteh, Weitao Yang, and Robert G. Parr. Physical review B 37, no. 2 (1988): 785.. For the CAM-B3LYP functional: Yanai, Takeshi, David P. Tew, and Nicholas C. Handy. Chemical physics letters 393, no. 1-3 (2004): 51-57.

3. page 3, computational details

The long-range interaction can play an important role in the studied systems, because there are many conjugated double bonds. Can the authors discuss the effects of not adding the dispersion correction in the DFT calculations?

4. page 5, Table 1

This table does not make too much sense, because all dihedral angels tabulated here are equal or very close to 180 degrees.

5. page 6, Figure 3

The resolution of this figure is too low, and ratios in each subfigure is not proper. In addition, the texts in this figure is hard to recognize.

6. page 8, Table 2

Can the authors specify the method for calculating HOMO and LUMO energies in this table? Is that still B3LYP?

7. page 8, Table 2

It should be mentioned in the context that band gaps predicted by DFT are commonly underestimated. And the basis set used here is not converged, which introduces further errors, especially for LUMO energies.

7. page 8, Table 2

The columns for EHOMO and ELUMO can be removed, since they are just the negative of IP and EA.

Comments on the Quality of English Language

No major language issue found. But there are some typos.

Author Response

Response to Reviewer #2's comments

(Manuscript: ijms-3011105)

We are very much thankful to the reviewer for his deep and thorough review. We have revised our present research paper in light of his useful suggestions and comments. We hope our revision has improved the paper to a level of satisfaction. Number-wise answers to his specific comments and suggestions are as follows:

Comment 1;

- page 1, title Typo. “DF-DFT” should be “TD-DFT”.

Response; 

Thank you for the note; the title has the right abbreviation DT-DFT

Comment 2;

- page 3, computational details

Some references are missing. For the B3LYP functional: Lee, Chengteh, Weitao Yang, and Robert G. Parr. Physical review B 37, no. 2 (1988): 785. For the CAM-B3LYP functional: Yanai, Takeshi, David P. Tew, and Nicholas C. Handy. Chemical physics letters 393, no. 1-3 (2004): 51-57.

Response;

As suggested by the reviewer, the missing references have been cited. Ref no; 35 and 36.

Comment 3;

The long-range interaction can play an important role in the studied systems, because there are many conjugated double bonds. Can the authors discuss the effects of not adding the dispersion correction in the DFT calculations?

Response;

Based on the reviewer's advice, a discussion about the long-range interaction and the effects of not adding the dispersion correction to the DFT calculations has been added to the introduction. (Lines 146 to 155)

Comment 4;

page 5, Table 1- This table does not make too much sense, because all dihedral angels tabulated here are equal or very close to 180 degrees.

Response;

As per the reviewer's recommendation, Table 1 has been removed from the main body of the paper and moved to the supplementary materials as Table S1.

Comment 5;

page 6, Figure 3 The resolution of this figure is too low, and ratios in each subfigure is not proper. In addition, the texts in this figure are hard to recognize.

Response;

The required changes to the text and resolution have been made to the maximum extent possible.

Comment 6;

page 8, Table 2. Can the authors specify the method for calculating HOMO and LUMO energies in this table 2? Is that still B3LYP?

Response;

Yes, the method used for calculating HOMO and LUMO energies in this table 2 is still B3LYP and has been included in the table's description.

Comment 7;

page 8, Table 2. It should be mentioned in the context that band gaps predicted by DFT are commonly underestimated. And the basis set used here is not converged, which introduces further errors, especially for LUMO energies.

Response;

As advised by the reviewer, the common underestimation of calculated band gaps by the standard density functional theory (DFT) methods and the non-converged basis set issue were discussed. (Lines 208 to 212)

Comment 8;

page 8, Table 2. The columns for EHOMO and ELUMO can be removed, since they are just the negative of IP and EA

Response;

The columns for EHOMO and ELUMO have been removed from Table 2, as suggested by the reviewer.

Thanks in Advance

Round 2

Reviewer 2 Report

Comments and Suggestions for Authors

The authors have correctly addressed all my technical remarks. Further review is not needed.

Author Response

We are very much thankful to the reviewer for his deep and thorough review
There is no a point-by-point response, as the reviewer stated " Further review is not needed."